# Minimal Residual Disease in Multiple Myeloma: Past, Present, and Future

**DOI:** 10.3390/cancers15143687

**Published:** 2023-07-20

**Authors:** Alejandro Medina-Herrera, María Eugenia Sarasquete, Cristina Jiménez, Noemí Puig, Ramón García-Sanz

**Affiliations:** Departament of Hematology, University Hospital of Salamanca (HUSA/IBSAL), CIBERONC, CIC-IBMCC (USAL-CSIC), 37007 Salamanca, Spain; amedinah@saludcastillayleon.es (A.M.-H.); jscris@usal.es (C.J.); npuig@saludcastillayleon.es (N.P.); rgarcias@usal.es (R.G.-S.)

**Keywords:** MRD, multiple myeloma, next-generation sequencing, flow cytometry, PET-CT, PCR

## Abstract

**Simple Summary:**

The assessment of responses is critical in patients diagnosed with multiple myeloma. Nowadays, one of the most informative parameters to discriminate responses to treatment and prognosis is minimal residual disease (MRD). Several strategies may be used to detect and quantify MRD; some of them have been widely used and standardized, but we can find additional strategies lacking such an extensive validation process. Here, we present a summary of the current state of the art of MRD detection in multiple myeloma and future directions in the field.

**Abstract:**

Responses to treatment have improved over the last decades for patients with multiple myeloma. This is a consequence of the introduction of new drugs that have been successfully combined in different clinical contexts: newly diagnosed, transplant-eligible or ineligible patients, as well as in the relapsed/refractory setting. However, a great proportion of patients continue to relapse, even those achieving complete response, which underlines the need for updated response criteria. In 2014, the international myeloma working group established new levels of response, prompting the evaluation of minimal residual disease (MRD) for those patients already in complete or stringent complete response as defined by conventional serological assessments: the absence of tumor plasma cells in 100,000 total cells or more define molecular and immunophenotypic responses by next-generation sequencing and flow cytometry, respectively. In this review, we describe all the potential methods that may be used for MRD detection based on the evidence found in the literature, paying special attention to their advantages and pitfalls from a critical perspective.

## 1. Prognostic and Clinical Relevance of Minimal Residual Disease in Multiple Myeloma

### 1.1. Definition of Minimal Residual Disease

Treatment of multiple myeloma (MM) has substantially advanced thanks to new drugs that have greatly improved the depth of response and the survival rate [1,2,3]. Treatment of MM includes high-dose therapy, proteasome inhibitors (bortezomib, carfilzomib, ixazomib), immunomodulatory drugs (thalidomide, lenalidomide, pomalidomide, iberdomide), and monoclonal antibodies (daratumumab, isatuximab, elotuzumab). Most recently, new targeted therapies such as antibody–drug conjugates, BCL2 inhibitors (venetoclax), XPO1 inhibitors (selinexor), CAR T cells, and bispecific antibodies have been also introduced [4]. Using modern drug combinations, most patients achieve complete response (CR). Nevertheless, many of these patients relapse [3], suggesting that a small but clinically relevant population of myeloma cells, known as minimal residual disease (MRD), persists. Given that is challenging to define what is ‘minimal’, in other diseases the term has been replaced by ‘measurable’ [5], although this concept has not been widely accepted in MM [6]. Conventional flow cytometry and molecular techniques such as quantitative PCR (qPCR) have been traditionally employed for the assessment of MRD in MM. However, deep responses achieved with new therapies require a higher level of sensitivity to detect residual cells. Next Generation Flow (NGF) and Next Generation Sequencing (NGS), together with digital PCR (ddPCR), mass spectrometry, and imaging techniques, have been developed to provide such sensitivity for MRD assessment.

The broad clinical advantages of MRD detection are perfectly enumerated in a large meta-analysis by Munshi et al. [7]. Here, the authors show how the MRD status impacts the outcome of patients, irrespective of serological responses (Figure 1). It retains its independent prognostic value in different clinical contexts, being predictive irrespective of the disease setting (newly diagnosed, relapsed/refractory, transplant-eligible or ineligible patients), the sensitivity thresholds, the cytogenetic risk, or the treatment scheme. Therefore, MRD fulfills most of the requisites to become a valid prognostic and predictive marker.

### 1.2. Baseline and Post-Treatment Outcome Predictors

Survival of MM patients has notably improved over the last 20 years, but a high heterogeneity between patients is still present. Thus, the search for baseline biomarkers capable of predicting progression-free survival (PFS) and overall survival (OS) is still key in myeloma [9]. Risk factors in MM are related to the patient’s characteristics, the tumor burden, the biology of the neoplastic plasma cell (PC), or a combination of these factors. Some of the most important risk factors have been included in Table 1. Most recent studies have shown that the major prognostic factors in MM are age and cytogenetics at diagnosis, together with the response to treatment during the follow up [9,10,11].

In 2005, the IMWG published the international staging system (ISS) based on serum β2-microglobulin and albumin [14]. In 2015, this system was updated by including serum lactate dehydrogenase (LDH) and cytogenetics into the revised ISS (R-ISS) [15]. The current consensus of the IMWG considers 17p deletions and translocations involving *IGH* and *MMSET*/*FGFR3* or *MAF* [t(4;14) and t(14;16), respectively] as high-risk alterations [11,18], while other patients with none of these fall in the standard-risk category. The R-ISS distinguishes three groups of patients with different 5-year PFS and OS rates, but it is still limited, and other prognostic alterations have been identified, but they are still not considered by the IMWG criteria. This has been the case of 1q gains or amplifications, 1p deletions, chromosome 21 trisomies, and *TP53* somatic mutations, all of them conferring poor outcomes, while chromosome 3 and 5 trisomies can predict better survival rates [16,18,19,20,21]. ‘Double-hit’ alterations (biallelic *TP53* aberrations and ISS III + 1q amplification) are not yet included in the R-ISS either, though they portend very poor outcomes [17]. Recently, a second revision of the R-ISS (R2-ISS) has included 1q gains or amplifications and excluded t(14;16) [16].

In addition to prognostic factors identified at diagnosis, response to treatment is the other most relevant predictor of patients’ outcomes.

### 1.3. Assessment of Treatment Responses in Multiple Myeloma Patients

M-protein levels in serum and/or urine form the basis of assessing response to therapy and progression in MM. The M-proteins consist on either intact or fragmented monoclonal antibodies secreted by tumor PC, so their detection is used as a surrogate marker to measure the abundance of PC. After initial proposals were produced by some groups [22,23], the first global consensus for response evaluation in MM was proposed by the European blood and marrow transplantation (EBMT) group that considered CR when meeting the following criteria: monoclonal protein disappearance in serum and urine by immunofixation, resolution of any soft-tissue plasmacytomas, and reduction of bone marrow (BM) PC (BMPC) to less than 5% [24]. CR and longer PFS and OS are strongly associated [25,26,27], but patients who achieve CR still progress even when maintenance therapy is continuously administered. How these patients should be monitored has been a matter of debate for a long time. Accordingly, the international myeloma working group (IMWG) introduced new response categories, very good partial response (VGPR) and stringent CR (sCR), as the best serological response that incorporated the normalization of serum free light-chains and the absence of clonal BMPC as additional requirements to the standard CR criteria [28]. Although CR and sCR correlated with improved PFS and OS rates in various studies [26,29], the vast majority of patients achieving deep responses relapse, reflecting the persistence of residual cells that remained undetected by conventional techniques and making clear that the achievement of responses could be enhanced: the role of serum free light-chains in patients already in CR was questioned [30], and sustained CR was not different to other sustained but inferior responses [31] in terms of survival. Therefore, in 2016, the IMWG urged the incorporation of new methods that are able to overcome the limit of detection of standard approaches to detect MRD. Immunophenotypic and molecular CR were defined [32] with the recommendation of introducing next-generation methods for MRD assessment into clinical trials.

## 2. Alternatives for the Detection of Minimal Residual Disease

### 2.1. Immunophenotypic Approaches

#### 2.1.1. Molecular Basis, Potential Targets, and Technical Considerations

The traditional immunophenotypic profile of pathologic plasma cells (PPCs) includes high expression of CD38 and CD138. Both are traditionally combined to identify PC in MM [33]. Other markers are aberrantly expressed in the PPC: CD19, which is usually absent; CD27, CD45, and CD81 generally have lower or negative expression compared to normal/reactive PC; and CD56, CD28, and CD117 are overexpressed in 60–75%, 15–45%, and 30–32% of cases, respectively [34]. It is also useful to incorporate markers of clonality such as cytoplasmic κ and λ light chains.

There is a great diversity of flow-based protocols to detect MRD in MM, which in turn means that a considerably high level of variability in the report of results exists. The number of colors that can be analyzed by the cytometer, the source of monoclonal antibodies, the selection of fluorochromes, and the total number of acquired events in each assay vary between laboratories. In the same line, detection parameters, sample processing, and data analysis are also diverse and lead to disparities in the final reports [34]. All of these issues underlined the need for stringent standardization of multiparametric flow cytometry (MFC) protocols [33,35,36]. The EuroFlow group has helped in the design and validation of monoclonal antibody panels and guidelines for interpretation, as well as in the optimization of tools for semi-automated analyses [37].

#### 2.1.2. Traditional and Next-Generation Flow Cytometry Approaches

MRD assessment by MFC in MM has proven to be very informative in numerous studies [38,39,40,41,42]. The applicability of MFC methods extends to almost all patients, and the time needed to process and analyze samples is relatively short. Despite these advantages, older 4- and 6-color flow cytometers had two major drawbacks that placed MFC behind molecular methods to detect MRD: first, data acquisition was limited to 100–500 K events, restricting its sensitivity to 10^−^^4^; in addition, the number of monoclonal antibodies per panel was also limited, making the identification of aberrant phenotypes much harder. The introduction of second- and third-generation flow cytometers (≥8 colors) and the possibility to acquire a higher number of events (5–20 million) have raised the limit of detection to the current 10^−^^6^, although standardization is still a key requirement [36,43,44].

In 2017, the EuroFlow group designed, optimized, and validated a monoclonal antibody panel for MRD monitoring in MM, together with a new sample processing protocol that allows for the analysis of a higher number of cells in a cost-effective manner. This approach, termed “next-generation flow cytometry” (NGF), combines the analysis of 10 markers (CD38, CD138, CD45, CD19, CD27, CD56, CD81, CD117, cytoplasmic Igκ, cytoplasmic Igλ) in two independent tubes analyzed by 8-color MFC [45]. The markers included in the panel allowed for the distinction between normal and clonal PCs and were also useful for the evaluation of other cell compartments in the BM. For instance, it allows the identification of sample hemodilution, a phenomenon that negatively affects the results of immunophenotypic and molecular studies [46]. In this regard, the Spanish myeloma group (‘Grupo Español de Mieloma’, GEM) has recently proposed additional guidelines to overcome such problem [47]. The EuroFlow’s NGF panel reaches a maximum sensitivity of 2 × 10^−^^6^ when at least 10 million events are analyzed, and it is currently recommended by the IMWG as a reference immunophenotypic method for the evaluation of MRD in MM [32].

Other groups have developed alternative strategies, with a wide variation in the selected monoclonal antibodies, fluorochromes, or sample pre-processing protocols, all of them reaching the minimum recommended sensitivity of 10^−^^5^. The Memorial Sloan Kettering Cancer Center group designed a single-tube, 10-color panel with a high concordance (98%) with the EuroFlow’s solution [48]; the Japanese group has developed their own 10-color panel achieving similar results [49]. Another 10-color, two-tube strategy was proposed by the Australian group that eliminates several washing steps, thus reducing the total length of the experiment [50]. The German group has also published their design of a single-tube, 10-color antibody panel including two additional markers (CD200 and CD28) highly positive in MM patients [51].

#### 2.1.3. Evidence

Several clinical trials have incorporated conventional MFC for MRD monitoring in MM, though only one has so far reported results using NGF (Table 2). The GEM group explored the use of conventional MFC in 162 newly diagnosed, transplant-ineligible patients treated in the GEM2010MAS65 trial (NCT01237249), achieving a limit of detection of 10^−^^5^ [43]. They described how MRD negativity was associated with an improved outcome irrespective of other relevant clinical parameters. In the GEM2012MENOS65 trial (NCT01916252) [52] evaluating the combination of bortezomib, lenalidomide, and dexamethasone (VRd) for induction and consolidation, as well as transplantation, in newly diagnosed, transplant-eligible patients, the EuroFlow’s NGF protocol was incorporated to evaluate MRD post induction, transplantation, and consolidation. In this context, achieving negative MRD demonstrated an 82% reduction in the risk of progression and 88% reduction in the risk of death, with a survival advantage observed even in high-risk patients [53]. In a second part of the study, the GEM2014MAIN trial (NCT02406144), the addition of 24 months of maintenance increased the rate of MRD negativity by 17% as compared to the end of consolidation [53]. In the phase III EMN02/HO95 study, patients received high-dose melphalan versus bortezomib, melphalan, and prednisone (VMP) during induction, followed by VRd versus no consolidation, and lenalidomide maintenance (NCT01208766). In the MRD sub-study performed by MFC with 10^−^^4^–10^−^^5^ sensitivity, 5-year PFS rates were 66% and 31% for MRD-negative and -positive patients, respectively. While MRD rates did not differ with or without consolidation (Table 2), 42% of patients turned MRD-negative after lenalidomide exposure in the maintenance phase [54]. In the phase III Myeloma XI (NCT01554852), MRD was serially assessed by MFC (median sensitivity: 4·10^−^^5^) following ASCT [55]. Those patients with sustained MRD negativity or MRD conversion from positive to negative over the first 9 months from ASCT had the longest PFS and OS rates. In addition, when patients were randomly assigned to lenalidomide maintenance or no maintenance post ASCT, a higher probability of MRD conversion from positive to negative was observed for those patients in the lenalidomide arm (30% vs. 17%), demonstrating the usefulness of continuous therapy. Very recently, Benjamin Diamond and colleagues have also described how sustained MRD negativity for two years during maintenance is associated with a lower risk of progression, compared to MRD persistence or conversion from MRD-negative to MRD-positive [56].

### 2.2. Molecular Approaches

#### 2.2.1. Molecular Basis, Potential Targets, and Technical Considerations

Multiple myeloma is an extremely heterogeneous disease in terms of genomic lesions [58]. Approximately half of patients are hyperdiploid, showing trisomies of odd chromosomes; the other half are characterized by the presence of translocations involving oncogenes that frequently juxtapose to the enhancer region of immunoglobulin (Ig) genes resulting in their upregulation. In addition, single-nucleotide variants (SNVs), copy number variants (CNVs), and indels are present in the major proportion of patients, affecting a wide range of pathways [58,59,60]. None of these alterations are useful for MRD evaluation in myeloma because they do not provide a specific marker for each patient, and clonal evolution shapes the genomic landscape of the disease over time.

Monoclonal rearrangements of *IGHV*, *IGHD*, and *IGHJ* genes in the three Ig loci (*IGH*, *IGK*, and *IGL*) provide highly specific V(D)J combinations for each B cell. This initial variability is increased by random insertions and deletions of nucleotides in the junction region, as well as by the introduction of somatic mutations during the antigen-dependent maturation steps. This processes result in a unique Ig sequence for each cell that can be used as stable biomarker for clonality detection in B-cell malignancies [61,62,63]. Detection of tumor-specific Ig sequences at diagnosis is, therefore, critical; using standardized approaches such as Sanger sequencing or next-generation sequencing, these targets can be detected in >90% of patients with B-cell lymphoid malignancies. Nonetheless, its applicability varies among different diseases. For instance, using leader or FR1 primers in patients with chronic lymphocytic leukemia (CLL) the applicability is virtually 100%, given the high tumor burden in peripheral blood and the relatively low somatic hypermutation rates in the Ig rearrangements. In contrast, multiple myeloma shows a patchy distribution of BMPC, with a high somatic mutation rate and frequent polyclonal background; all these features make amplification and interpretation of results more challenging, so the applicability of sequencing approaches in MM is 80–90% [64].

The first approach targeting junction regions consisted in designing unique combinations of primers and probes (allele-specific oligonucleotides, ASO) for each patient followed by their amplification in a nested PCR or, more recently, real-time quantitative PCR (qPCR) [65,66,67]. Both methods reach sensitivity thresholds of 10^−^^5^–10^−^^6^ (detecting up to one tumor cell in 100,000−1,000,000 normal cells), but they are time-consuming and their applicability is highly variable. In contrast, novel molecular strategies (ddPCR, NGS) seem more appealing as they outperform traditional approaches.

#### 2.2.2. Allele-Specific Oligonucleotide Quantitative PCR

In MM patients, monoclonal rearrangements are the traditional markers for MRD detection. This strategy may be applied in a qualitative or a quantitative PCR approach; however, due to the lack of quantitative power and increased risk of contamination, no efforts have been made so far in the qualitative setting, with real-time quantitative allelic-specific oligonucleotide qPCR (ASOqPCR) being the gold standard. This approach achieves a sensitivity threshold of 10^−^^5–^10^−^^6^, but it relies on the nature of the V(D)J rearrangement (whether it is the heavy-chain or light-chain rearrangement), the length of the junction region, the specificity of primers and probes, and the amount of DNA available.

MRD testing by ASOqPCR includes two major steps: the first one consists of the identification of clonal V(D)J rearrangements at diagnosis to design a unique combination of primers and probes; the second one is the amplification and detection of those sequences by qPCR. The first step is complex because it requires sequencing analysis of the hypervariable region before the designing of clone-specific primers and probes for each patient. For this purpose, the BIOMED-2 group (now Euroclonality) standardized several multiplex primers for *IGH*, *IGK*, and *IGL* clonality testing [68]. After the identification of clonal peaks by GeneScanning or heteroduplex analysis, PCR products are then further sequenced by Sanger sequencing to identify the patient-specific sequence of the corresponding tumor PC (the complementarity-determining region 3, CDR3) to design the specific primer/probe combination.

Despite their ongoing development since discovery, PCR-based methods still present some limitations. One of the most relevant issues is the rate of marker identification failure, which depends on the molecular target, the availability of consensus primers, the tumor infiltration of the sample, and the biology of the disease. Thus, while B-cell acute lymphoblastic leukemia, mantle-cell lymphoma, and CLL are successfully monitored by ASOqPCR in ≥90% of cases, in MM, it has shown applicability rates of 40–75% [66,69]. This is mainly due to the presence of high rates of somatic hypermutation, frequent in post-germinal center hematological disorders, which hampers both steps of ASOqPCR—i.e., the identification of the tumor markers at diagnosis and the correct quantification of the tumor burden in follow up samples—due to inappropriate primer annealing. The limited applicability of MM can be improved including additional targets [69,70]. However, from a methodological perspective, labor intensiveness is a major concern, as well as the need to build a standard curve of serial dilutions to assess a correct quantitation of targets and the adequate design of patient-specific primers, which requires well-trained laboratory staff and is indeed time-consuming. There are also some drawbacks causing false-negative results that should be mentioned: BM hemodilution, the patchy distribution of malignant PC within the BM, or low infiltration rates. All these, together with extramedullary disease, represent common pitfalls to those techniques relying on the analysis of BM aspirates, which may not be representative of the tumor.

In contrast, the ASOqPCR approach has several advantages: it does not require fresh samples, and a rigorous standardization program was carried out within the EuroMRD group, a division of the European Scientific Foundation for Laboratory Hemato-Oncology (ESLHO) Consortium [65,71]. This has led to the international harmonization of MRD assessments and the introduction of qPCR in several clinical trials for other hematological malignancies [72,73,74]. Despite these potential advantages, problems mentioned above have currently displaced the use of qPCR in favor of more recent technologies in the case of MM.

#### 2.2.3. Evidence

Bakkus et al. analyzed the usefulness of ASOqPCR to detect MRD 3–6 months post ASCT in 67 MM patients [75]. Patients were classified according to the MRD status, and those with detectable disease displayed a shorter time to relapse.

Later on, in the GEMM2000 trial (NCT00560053), 32 patients achieving at least near-CR after ASCT were included for MRD studies by ASOqPCR that were successfully carried out in 24 of them (75% of applicability). The low-MRD group displayed a significantly longer PFS than the high-MRD group (34 months vs. 15 months, *p* = 0.042) [66]. The same group compared the MRD status in a larger series of 170 patients included in two clinical trials achieving at least partial response at the end of induction or after ASCT [69]. Four-color MFC and ASOqPCR were employed and compared, achieving a significant correlation in the quantification of MRD levels (r = 0.881; *p* < 0.001). However, more than half of the patients could not be studied by ASOqPCR due to technical limitations (lack of molecular marker, unsuccessful sequencing, or suboptimal ASOqPCR performance). In light of the high proportion of cases that failed to estimate the MRD level, two modifications were considered to overcome this issue: the first one was using *IGH* DJ [76] and *IGK* rearrangements [70] as alternative molecular targets, given the lower impact of somatic hypermutation in these regions. This improved the applicability by 9%. A second proposal consisted of sorting CD138+ cells from BM samples before DNA extraction, which consistently increased the detection rate from 60% to 96% [77].

Furthermore, Korthals and colleagues [78] analyzed the role of MRD levels before ASCT in 53 MM patients receiving high-dose chemotherapy followed by ASCT. A 0.2% threshold was used for the stratification of patients according to MRD; being classified into the low-MRD group was an independent predictor for both increased event-free survival (EFS; median 35 months vs. 20 months, *p* = 0.04) and OS (median: 70 months vs. 45 months, *p* = 0.04). Ladetto et al. analyzed the effect of the combination of bortezomib, thalidomide, and dexamethasone (VTD) for consolidation in 39 patients achieving ≥ VGPR after ASCT, showing that molecular remission increased from 3% post ASCT to 18% post consolidation [79].

#### 2.2.4. Next-Generation Sequencing

NGS stands as the alternative to qPCR, and its potential to overcome some of its major disadvantages made it a promising tool for MRD evaluation in MM. For MRD analysis, both amplicon- and targeted-capture-based sequencing methods have been applied, although the first one is often preferred due to its lower cost, increased sensitivity, and more user-friendly basis.

Several protocols have been developed to sequence Ig receptors using this technique, consisting of a consensus multiplex PCR where combinations of *IGHV* and *IGHJ* primers allow the amplification of (virtually) all rearrangements present in the sample. The introduction of adaptor and index sequences may be performed in the same PCR reaction or in multiple PCR steps. The vast majority of MRD studies using NGS have been carried out with the ClonoSEQ strategy (Adaptive Biotechnologies, Seattle, WA, USA), but other commercial or in-house protocols have been described [80,81,82,83]. At baseline, NGS allows clonality detection. Typically, a threshold of 5% of identical sequences among all the other detected sequences is used to define clonality [84], although more stringent criteria have been proposed [64,85]. After therapy, resistant tumor cells can be traced using the patient-specific clonotype with superior specificity and sensitivity compared to qPCR, and tumor burden can be quantified using a spike-in strategy [86,87].

#### 2.2.5. Evidences

The introduction of highly sensitive techniques to detect MRD has refined the detection of patients with an increased risk of progression or death, including publications from the Spanish [88] and the French [89] myeloma groups, where NGS was used. These studies demonstrated improved PFS and OS rates for MRD-negative patients, highlighting the superiority of MRD detection over conventional response assessment as an outcome predictor, even when complete or stringent complete responses are achieved. Moreover, they also showed that MRD is able to stratify patients with different outcomes not only when it is used as a qualitative biomarker (positive or negative) but also when different quantitative levels are considered. Finally, the French report underlined the potential benefit of achieving MRD negativity for patients carrying high-risk cytogenetic alterations, such as t(4;14) translocations, a finding that has been also replicated in recent publications [90,91].

All these evidences led the IMWG to recommend the use of next-generation sequencing and next-generation flow as the preferential molecular and immunophenotypic approaches to detect MRD in the BM of MM patients [32], prompting the inclusion of such methods in prospective clinical trials that could ultimately be used to standardize and validate criteria for the interpretation of MRD results in the future [92]. Therefore, in the last years, MRD evaluation by NGS has been extensively used in the context of myeloma trials (Table 3) evaluating new drug combinations.

Results from some of these studies were submitted for inspection by regulatory agencies and, in 2018, the ClonoSEQ’s NGS assay was cleared by the FDA as the first method to evaluate MRD in MM, supporting its role as a major predictive factor, as has already been shown in several meta-analyses [3,93].

Two major phase 3 randomized trials testing quadruplets are currently active and consider the evaluation of MRD by NGS: Dara–VRd in the CEPHEUS (NCT03652064) and Isatuximab–VRd in the IMROZ (NCT03319667). Trials investigating the use of CAR T cells and bispecific antibodies in relapsed/refractory patients have also demonstrated the usefulness of NGS methods to evaluate responses through MRD quantitation. CARTITUDE-1 evaluated ciltacabtagene autoleucel in triple-class previously exposed patients (PI, IMiD and anti-CD38). The updated follow-up showed impressive results, with 82.5% of patients (N = 80/97) achieving sCR and 57.7% of MRD-negative patients (N = 56/97) at 10^−^^5^; 68% (N = 34/50) and 55% (N = 24/44) of cases had MRD-negative status sustained for at least 6 or 12 months, respectively [94]. In the MajesTEC-1 trial (NCT03145181 and NCT04557098) for patients with at least three prior lines of therapy, 65/165 patients (39.4%) achieved CR or better, and 44 (26.7%) showed MRD negativity by NGS at 10^−^^5^ with teclistamab [95]. However, MRD dynamics seems to be different after using these new drugs, so more information about MRD is still required in this setting.

#### 2.2.6. Digital PCR

Droplet digital PCR (ddPCR) brings a new level of precision in quantifying nucleic acids, and it is a promising tool for MRD monitoring, increasing the chance of identifying molecular targets (such as *IGH* rearrangements, fusion genes, or mutations) to be followed after therapy. It is based on three compass points: (1) target compartmentalization, (2) end-point PCR, and (3) Poisson statistics. Compared with traditional qPCR approaches, ddPCR has already been foreshown as more precise, better at detecting rare genetic events, and less susceptible to inhibitors. In addition, ddPCR presents several practical advantages over qPCR, such as the lack of standard curves. Thus, this method may overcome pitfalls associated with fluctuations in reaction efficiency and makes MRD detection possible for all those patients in which generating a standard curve is not feasible.

ddPCR has recently been adopted for MRD measurement in different hematologic malignancies, such as MM, mantle-cell lymphoma, follicular lymphoma, and acute lymphoblastic leukemia, in the context of European MRD working groups. It has shown an excellent correlation with MRD measured by standardized qPCR methodologies, with a high percentage of concordant results. In fact, most discordances are usually detected in samples with low MRD levels, in which ddPCR is able to identify and quantify residual disease thanks to its higher sensitivity when qPCR does not [103].

#### 2.2.7. Evidence

Drandi et al. performed MRD studies in patients with MM, follicular lymphoma, and mantle-cell lymphoma, comparing the performance of traditional qPCR and ddPCR approaches. Their results showed a good concordance between both techniques in all three pathologies, with ddPCR being more applicable and less labor intensive than qPCR. Sensitivity, accuracy, and reproducibility of ddPCR were at least comparable to qPCR and avoided pitfalls related to the standard curve [104].

Thus, ddPCR may be considered as an alternative tool for MRD assessment in lymphoid malignancies including MM patients, although its usefulness needs to be conclusively documented in the context of prospective clinical trials. Currently, ongoing efforts are being conducted by the EuroMRD group to make ddPCR results reproducible across different laboratories, with standardized guidelines for its interpretation.

### 2.3. Overall Comparison of Contemporary MRD Strategies

One of the most important features of both NGS and 8–10 color MFC is their superior sensitivity, commonly set in the range of 10^−^^5^–10^−^^6^ [45,84,105,106]. In a study conducted by Ladetto and colleagues, MRD evaluation by ASOqPCR and NGS were compared [107]. Both techniques were concordant in 96% of patients and sensitivity was set as 10^−^^5^ for both methods. NGS did not require patient-specific oligonucleotides and standard curves. Since sensitivity is limited by the total number of input cells or DNA equivalents, a recent publication using NGS has even achieved a threshold of 10^−^^7^ using more than 100 µg of DNA per patient [108]. Nonetheless, this seems to be hardly implemented in the clinical routine, because a high volume of BM blood is required and the need to test multiple replicates would make this approach too expensive.

The applicability of NGS is high, in line with all PCR-based methods. Despite the high incidence of somatic mutations targeting Ig rearrangements, using different primer sets allows the identification of clonotypic sequences in at least 90% of cases. Using NGF, only those cases where the aberrant phenotype is difficult to detect may represent a challenge, but these are extremely rare, making it so that virtually all patients are susceptible to be analyzed.

A major advantage for NGS is the possibility of storing DNA samples, avoiding the need for fresh samples and a short processing time demanded by flow-based methods. NGS could be used in a similar way to NGF in order to monitor the normal immune compartment over time, as it has been used in recent publications showing the importance of polyclonal recovery after therapy [109,110,111,112].

However, and in line with qPCR, NGS cannot detect hemodilution in BM samples, and both molecular methods still rely on the identification of clonotypic rearrangements at baseline. On the contrary, NGF provides a source for intrinsic quality control checks based on the enumeration of mast cells, B-cell precursors, and nucleated red blood cells proposed by the GEM and EuroFlow groups [47]. BM-based methods, either sequencing or flow, lack the ability to detect extramedullary relapses, which—together with the patchy distribution of myeloma cells in the BM—are critical sources of error; therefore, false-negative cases cannot be avoided. Thus, it is crucial to get the first marrow pull to have the most representative sample available for MRD analyses. In fact, two recent publications suggested the combinatorial use of flow and imaging for improved MRD monitoring [113,114] in and outside the BM; focal lesions were observed in CR patients achieving MRD negativity after first-line (12%) or multiple-line therapy (50%), and the same approach could be useful coupling imaging with NGS. Another promising possibility is the use of liquid biopsies to monitor circulating plasma cells without the need to perform BM biopsies, a field that has been mostly explored by flow cytometry with important prognostic implications [115,116,117,118].

NGS quantification depends on a comparison between the amplification of tumor and spike-in DNA controls, which may not be linear. On this matter, there are ongoing efforts in order to set reliable control references that could be used to quantitate MRD using high-throughput sequencing technologies [119]. The availability represents another drawback for currently accepted technologies: to date, there is only one validated method for each strategy. The EuroFlow’s NGF solution requires the use of specific reagents, sample pre-processing protocols (bulk lysis), and detection parameters. The ClonoSeq’s NGS strategy is a method accessible only by sending samples to the core facilities in the USA, and its cost makes it unaffordable for most laboratories. Finally, while flow results are shortly provided, turnaround time for NGS is longer, and this would make its implementation in the routine clinical setting more difficult. A comparison between molecular and flow approaches is shown in Table 4.

## 3. Future Perspectives

### 3.1. Liquid Biopsy: Circulating Tumor DNA and Circulating Tumor Cells

Liquid biopsy is a non-invasive strategy for disease monitoring through the analysis of circulating tumor DNA (ctDNA) or circulating tumor plasma cells (CTCs). This approach is becoming a promising non-invasive tool, notably for monitoring response to treatment in lymphomas [120,121,122]. Moreover, the development of these blood-based MRD strategies is crucial to overcome pitfalls related to BM samples. Detecting ctDNA or CTCs is of particular relevance in MM as both sources better represent the multifocal “patchy” nature of the disease, rather than relying on aspiration at a single marrow site to reflect the complete cancer milieu, which is not realistic. In addition, recurrent and frequent sampling of peripheral blood is feasible and, as opposed to BM, painless. Although its clinical utility is still under investigation, new emergent data is becoming available in the context of plasma cell disorders (PCDs) [123].

The evaluation of ctDNA for mutational characterization and monitoring of disease burden has recently been described in a few studies. ctDNA levels captures tumor heterogeneity and tumor kinetics, although results are controversial when compared to BM, since a lack of correlation between mutations and/or MRD rates has been detected [124,125,126,127,128,129]. Monitoring patients with liquid biopsies becomes specifically useful for patients with oligosecretory or non-secretory MM, and in patients with extramedullary MM (EM-MM), where conventional markers of tumor burden are not adequate and a BM biopsy is not always plausible, respectively. In such cases, the concordance between ctDNA and plasmacytomas is apparently higher, thereby suggesting that ctDNA is a promising surrogate material for the mutational characterization of EM-MM, particularly when plasmacytomas are not accessible [130,131].

CTCs, identified with sensitive MFC methods, are found in more than half of the patients diagnosed with monoclonal gammopathy of undetermined significance, as well as in all newly diagnosed smoldering and symptomatic MM patients, but are rarely identified in patients with solitary plasmacytoma [115]. Their detection is currently recognized as a key hallmark of aggressive disease and, given their strong association with survival, the quantification of CTCs may soon replace conventional quantification of BMPC in the next-to-come staging systems for plasma cell leukemia [132], symptomatic MM [117,118], and precursor stages [133]. As with ctDNA, genome sequencing of CTCs enables minimally invasive molecular profiling of MM in its real spectrum, without the limitations associated with the irregular distribution of tumor cells in the BM [134,135,136]. However, technologies required for such characterization are available only in selected laboratories, which limits their applicability.

Overall, liquid biopsies may provide a dynamic and comprehensive picture of the genomic landscape in MM and, even more, a non-invasive approach to monitor tumor burden. However, these methods are still novel and demand further research, especially when comparing results with matched BM assessments [137,138]. Therefore, the implementation of liquid biopsies for MM requires validation and harmonization of the assays [123].

### 3.2. Imaging Techniques

One of the hallmarks of MM is the presence of bone lesions occurring in virtually all patients at diagnosis or during the course of the disease, representing a major cause of morbidity and mortality. Bone disease was traditionally assessed by whole-body X-ray, which represented for a long time one the main criteria used to define the start of therapy, but its sensitivity was not always sufficient. The introduction of whole-body low-dose computed tomography showed that approximately 25% of patients with a negative skeletal survey had osteolytic lesions [139]. Positron-emission tomography and computed tomography with ^18^F-fluorodeoxyglucose (^18^F-FDG PET/CT), as well as whole-body magnetic resonance imaging (WB MRI), are both sensitive approaches to detect focal lesions, extramedullary disease, and a diffuse tumor infiltration pattern in the BM [140]. In the context of assessing responses to treatment, several cohorts have been extensively evaluated using these approaches, showing the correlation of a negative result with good responses and long survival rates [141,142,143]. Accordingly, the last guidelines from the IMWG stablished PET/CT as the reference tool to define imaging + MRD negativity when it is coupled with NGS or MFC [32]. Since imaging captures the expansion of tumor cells at a whole-body level, distinguishing active from inactive lesions, perhaps it should be interpreted as a second layer of MRD.

Over the last decade, ^18^F-FDG PET/CT has been introduced for prognostication in several clinical trials [144,145,146], but it is not yet routinely implemented due to its cost, low availability, and standardization requirements [147,148]. Nowadays, how to use PET/CT results is controversial, since it is standardized only at the pre-maintenance stage in newly diagnosed, transplant-eligible patients [143,149]. Moreover, using ^18^F-FDG may lead to false-positive and false-negative results. To overcome this limitation, alternative markers (choline, methionine, thiotymidine, fluciclovine, and others) labeled with ^11^C, ^18^F, ^64^Cu, ^68^Ga, or ^89^Zr have been proposed [150,151,152,153].

MRI examination is a sensitive method to detect BM infiltration by MM cells before significant bone destruction is present. WB diffusion-weighted MRI (DWI) emerged as a promising option for response assessment with several advantages over PET/CT: it lacks exposure to radioactive agents; the sensitivity, including the identification of a patchy infiltration of abnormal PC in the BM, is better; and there is no impact of the level of hexokinase expression in tumor PC, a frequent cause of false-negative PET/CT results [154,155,156]. However, PET/CT has been more extensively evaluated in the literature compared to MRI, with only a few retrospective studies [142], and consensus standardization of WB MRI for response assessment in MM is relatively recent, because the sensitivity of MRI was lower prior to the application of the DWI modality [141]. A prospective head-to-head comparison of both technologies performed by the French group demonstrated that the normalization of MRI after three cycles of VRd and before maintenance was not predictive of PFS or OS. By contrast, PET/CT became normal after three cycles of VRd in 32% of the patients with a positive evaluation at baseline, and 30-month PFS was improved in this group (78.7% vs. 56.8%, respectively) [145]. Although these results were restricted to early responses, no new comparisons have been performed, so response evaluation by imaging techniques is performed preferentially by PET/CT, following the above written recommendation of the IMWG.

### 3.3. Mass Spectrometry

All MS methods have a similar basis for detecting M-proteins: the unique sequence of the three CDRs of the Ig. Since each PC encodes a specific Ig with a unique amino acid sequence, the resulting protein has a particular structure with a specific overall mass, which is accurately detected by MS. Two main MS approaches have been described so far by the IMWG [157], both taking the enrichment of each patient’s Igs as the starting point, but with different downstream detection and analysis of the target molecule. One of these methods divides the Ig into peptides specific to the CDR by enzymatic digestion (clonotypic peptide approach), while the other one chemically reduces and denatures Igs into heavy and light chains to measure the distribution of the LC mass (intact LC approach).

The clonotypic peptide approach is analytically very sensitive with a detection rate down to 0.001 g/L of M-protein [158]. However, it is not applicable in a subset of patients for whom the identification of tumor-specific peptides is not possible: clonotypic rearrangements contain framework regions that are not as diverse as CDRs, and once a potential clonotypic peptide is identified, it must be filtered with a reference proteomic database to assure its uniqueness. While this approach has been successfully used for MRD tracking and detection of therapeutic monoclonal antibodies (e.g., daratumumab, isatuximab, and others) [159,160,161], the intact LC approach is simpler and more practical. Here, the polyclonal repertoire of Ig LCs of each patient is analyzed independently for each one of the two LCs. A first version of the approach coupled MS with liquid chromatography and was termed ‘monoclonal immunoglobulin rapid accurate mass measurement’ (miRAMM), with a limit of quantitation of 0.05 g/L and a limit of detection of 0.01 g/L [162]. An improved version incorporated matrix-assisted laser desorption/ionization-time-of-flight (MALDI-TOF) to MS, thus eliminating the liquid chromatography step, reducing the turnaround time and increasing the sensitivity if an enrichment step is also performed. The intact LC MALDI-TOF MS assay, or ‘Mass Fix’, has been extensively validated at the Mayo Clinic with thousands of samples [163,164,165] and several comparisons with NGS or NGF unveils its prognostic value, with a complementary role to BM-based methods [166,167,168,169].

Besides these two approaches, the first results of a third MS version termed ‘quantitative immunoprecipitation mass spectrometry’ (QIP-MS) have been recently presented [170]. This assay enables the identification, quantification, and typing of complete and LC-only M-proteins at once. Preliminary results from the Spanish GEM group showed a strong association of MRD and a shorter survival detected by either QIP MS or NGF, with a high correlation between techniques [171]. However, more studies are needed to discern the best MS method and its clinical implications.

## 4. Open Questions in the Field of MRD

Despite recent advantages in the evaluation of MRD, a lack of information is still present, and important questions remain unanswered.

(a)Which patients should be evaluated?

The major proportion of studies evaluating MRD only consider patients in complete response, following the absence of detectable M-protein in serum and urine and reduced plasma cell counts in the BM. However, some studies have found a subset of patients achieving only VGPR after induction or transplantation but who are MRD negative [6,63]. This is probably a consequence of the dynamics of tumor cells and monoclonal proteins: malignant cells may be already eradicated while the monoclonal component is still detectable in serum or urine, although such discordances are sometimes sustained during long time periods. Such findings bring up what the optimal timing for MRD detection is and the relevance of sustained MRD negativity.

(b)What is the optimal time point for MRD evaluation?

In clinical trials evaluating previously untreated patients, MRD is usually monitored at the end of induction, following transplantation in fit patients (three months is the most common time point), and periodically thereafter. Several publications have stated that MRD is not a static parameter and thereby clinical decisions should be taken considering its fluctuations over time. What is more, for a given patient, a single MRD-negative result could be potentially achieved with several drug combinations. While this still has clinical implications, MRD disappearance may be transient, so only the most beneficial treatment scheme (and lethal for MM cells) will result in a sustained clearance of tumor cells.

Therefore, reporting sequential MRD negativity rates would be preferred in clinical trials as the best indicator of therapy effectiveness. In fact, the IMWG consensus paper recommended the confirmation of negativity a minimum of one year apart [32], although the definition of ‘sustained’ was randomly decided, as mentioned in the original article. Gu and colleagues [172] suggested that MRD should be performed post induction as well as 3 and 24 months after transplantation and described how patients with sustained MRD negativity for 2 years after the end of induction had similar outcomes to those becoming MRD negative within 24 months, but longer OS than patients with sustained MRD positivity or MRD resurgence, results similar to those reported in a phase II trial from the Memorial Sloan Kettering Cancer Center [56].

At the current time, we consider the combined analysis of ALCYONE and MAIA trials, as well as TOURMALINE-MM3 and -MM4 trials, as the most mature studies focusing on MRD interpretations. In the pooled analysis of MRD data obtained from MAIA and ALCYONE, the authors show the impact of sustained MRD (≥6 or ≥12 months) on patients’ prognosis compared to a single MRD assessment, and how daratumumab-containing regimens render the highest (single and sustained) MRD-negative rates, ameliorating the negative impact on PFS of MRD positivity and/or loss of MRD negativity [98]. Results from the TOURMALINE-MM3 and -MM4 trials also highlight the prognostic value of continuous assessment of MRD compared to a single time point, extending MRD analyses during maintenance [173]. Here, the authors found a 9.5% rate of MRD resurgence (from negative to positive) that predicted poor outcomes similar to patients with sustained positive MRD. Conversely, 5.1% of patients experienced MRD negativization and had similar outcomes to patients with sustained negative MRD.

Thus, a consensus definition of the term ‘sustained’ is primordial in order to define potential tailored treatment strategies and, ultimately, update the term ‘operational cure’. In light of recent publications, sustained negative MRD or MRD negativization would lead to improved outcomes and a lower risk of progression, while loss of MRD negativity mimics sustained MRD positivity, both robustly anticipating progression. Early intervention may represent an alternative in this situation that should be prospectively evaluated in clinical trials.

(c)How can we use MRD information?

Using MRD to make clinical decisions in MM is promising, and initial approaches in this line have already been explored. Two alternative pathways may be taken: treatment intensification or de-escalation for MRD positive and negative patients, respectively. Martínez-López and colleagues have recently suggested a clinical benefit in terms of prolonged PFS for those patients whose therapy was adapted based on MRD results by NGS [174]. The real usefulness of MRD-driven treatments will be ascertained in prospective clinical trials, some of them already ongoing for transplant-eligible patients [MASTER (NCT03224507), PERSEUS (NCT03710603), AURIGA (NCT03901963), or MIDAS (NCT04934475) trials]. In fact, results from the MASTER trial have been recently published, showing the benefit of therapy cessation when sustained MRD negativity is accomplished, although patients with ≥2 high-risk cytogenetic abnormalities had inferior PFS and OS than other patients, despite similar MRD negativity rates [175]. Additional biological characteristics of tumor cells may impact the prognostic usefulness of MRD (for instance, the transcriptional program or the epigenetic status of residual cells, or the composition of the BM microenvironment).

(d)Which strategy is the best?

With all the strategies that have been described in this article, choosing the optimal technique to monitor MM patients represents a real challenge. The ideal method for MRD monitoring should allow the identification of PPCs in a sensitive, prognostic, non-invasive, standardized, cost-effective, and pan-regional approach. Such a strategy currently seems unavailable, and, therefore, the most informative workflow would result from the combination of at least two methods, preferably in and outside the BM, one of them being standardized (NGF or NGS solutions as recommended in the last consensus response criteria from the IMWG) [32]. Nonetheless, we particularly envision a future replacement of BM studies with liquid biopsies for MRD as the most probable picture.

## 5. Conclusions

Detection of MRD in multiple myeloma represents an exciting field, with strong evidence supporting its prognostic role during the entire course of the disease, and a plethora of new alternatives under constant development to identify residual tumor cells in and outside the BM. The next steps will most probably unveil which are the most informative MRD alternatives, when to perform an MRD evaluation, and how to modulate treatment strategies based on MRD results. With all the upcoming evidence, individualized patient care in myeloma may be more realistic.

With this body of evidence, and based on our experience, we provide the following consensus recommendations:(1)MRD studies should be performed in the bone marrow, using validated and standardized procedures capable of assessing high sensitivity thresholds, ideally 10^−^^6^, which currently includes only NGF and NGS. Each institution may choose the most appropriate based on availability, expertise, and other aspects.(2)Bone-marrow-based MRD analyses should be performed using the first pull of the aspirate to prevent hemodilution. Evaluating MRD outside the bone marrow is an appealing and complementing option; incorporation of liquid biopsies and imaging techniques in prospective clinical trials would be very helpful to avoid invasive procedures, although more evidence is needed.(3)The evaluation of MRD should be performed in parallel with other clinical routine assessments at relevant time points, at least including post induction, post transplantation in candidate patients, post consolidation, post maintenance, and periodically during the subsequent follow-up. Single-time-point MRD may be prognostic and informative, but consecutive assessments in order to characterize MRD kinetics should be the goal.(4)If available, we recommend performing MRD studies for all patients diagnosed with multiple myeloma, in and outside of clinical trials. Results must be interpreted in the particular clinical–biological context of each patient and used for prognostic purposes. Interventional strategies based on MRD should be limited to clinical trials designed with that aim.

## Figures and Tables

**Figure 1 cancers-15-03687-f001:**
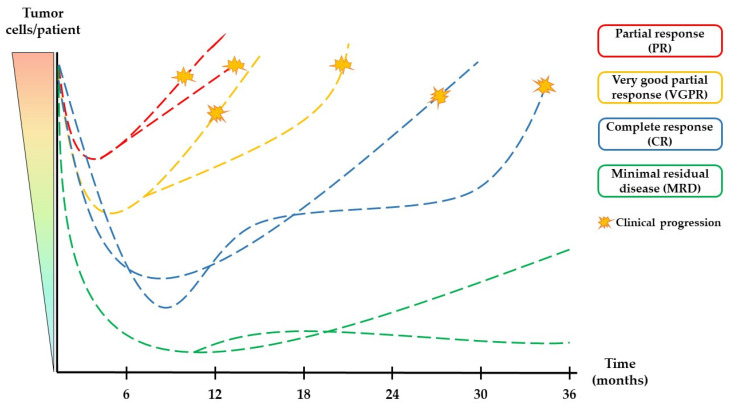
Levels of response in multiple myeloma. Serological methods provide enough sensitivity to detect partial or nearly complete disappearance of tumor cells by detection and quantification of the M-protein. However, even in patients achieving (stringent) complete responses, there is frequently still a minor population of tumor plasma cells (minimal residual disease, MRD) that are detectable only by highly sensitive methods in or outside the bone marrow. Therefore, the absence of detectable residual disease identifies a subset of patients with, in principle, longer progression-free and overall survival probabilities. The magnitude of reduction of the tumor burden is usually associated with the time required for progression [8], although many other factors—current and prior treatment regimens, risk markers associated with tumor biology, other co-morbidities, and more—play a role in this matter.

**Table 1 cancers-15-03687-t001:** Classical and novel risk markers in multiple myeloma. HR: high risk; ISS: international staging system; OS: overall survival; R-ISS: revised ISS; R2-ISS: second revision of the ISS.

Risk Marker	Prognostic Impact	References
*Classical*		
**Age**	Survival decrease with each decade of life	[12]
**Baseline HR cytogenetics**	OS < 3 years	[13]
**ISS, OS (months)**		[14]
I	62	
II	44	
III	29	
**R-ISS, 5-year OS rate**		[15]
I	82%	
II	62%	
III	40%	
*Novel*		
**R2-ISS, OS (months)**		[16]
I	NR	
II	109.2	
III	68.5	
IV	37.9	
**Double-hit, OS (months)**	20.7	[17]

**Table 2 cancers-15-03687-t002:** Phase III clinical trials incorporating MRD evaluation by MFC in MM. ASCT: autologous stem cell transplantation; Dara: daratumumab; d: dexamethasone; MRDneg: negative minimal residual disease; R: lenalidomide; T: thalidomide; V: bortezomib.

Study	Sensitivity (Median)	Treatment Algorithm	MRDneg Rates
*ASCT-eligible*			
**Myeloma XI**	4 × 10^−5^	ASCT + R vs.	65.6%
(NCT01554852) [55]		no maintenance	34.4%
**GEM2012MENOS65**	3 × 10^−6^	VRd + ASCT + VRd	50.2%
(NCT01916252) [52,53]			
**CASSIOPEIA**	10^−5^	Part 1: Dara − VTd + ASCT + Dara − VTd vs.	64%
(NCT02541383) [57]		VTd + ASCT + VTd	44%
	10^−5^	Part 2: Maintenance with Dara vs.	66%
		observation	55.2%
**EMN02/HO95**	10^−5^	Consolidation with VRd vs.	9.8%
(NCT01208766) [54]		No consolidation	8.2%

**Table 3 cancers-15-03687-t003:** Phase III clinical trials incorporating MRD evaluation by NGS in MM. ASCT: autologous stem cell transplantation; Dara: daratumumab; d: dexamethasone; Isa: isatuximab; K: carfilzomib; M: melphalan; MRDneg: minimal residual disease; P: prednisone; R: lenalidomide; T: thalidomide; V: bortezomib.

Study	Sensitivity (Median)	Treatment Algorithm	Mrdneg Rates
*ASCT-eligible*			
**IFM2009**	10^−6^	VRd, 8 cycles vs.	20%
(NCT01191060) [89]		VRd + ASCT	30%
**CASSIOPEIA**	10^−5^	Part 1: Dara − VTd + ASCT + Dara − VTd vs.	57%
(NCT02541383) [57]		VTd + ASCT + VTd	37%
	10^−6^	Part 2: Maintenance with Dara vs.	49.5%
		observation	36.7%
**GRIFFIN**	10^−5^	Dara − VRd + ASCT + Dara − VRd vs.	51%
(NCT02874742) [96]		VRd + ASCT + VRd	20.4%
*ASCT-non-eligible*			
**ALCYONE**	10^−5^	Dara − VMP vs.	22%
(NCT02195479) [97,98]		VMP	6%
**MAIA**	10^−5^	Dara − Rd vs.	24.2%
(NCT02252172) [98,99]		Rd	7.3%
*Relapsed/refractory*			
**CASTOR**	10^−5^	Dara − Vd vs.	15%
(NCT02136134) [100]		Vd	1.6%
**POLLUX**	10^−5^	Dara − Rd vs.	33.2%
(NCT02076009) [101]		Rd	6.7%
**IKEMA**	10^−5^	Isa–Kd vs.	29.6%
(NCT03275285) [102]		Kd	13%

**Table 4 cancers-15-03687-t004:** **Current molecular and immunophenotyping MRD approaches.** * The EuroClonality-EuroNGS consortium has also developed and validated a standardized protocol for B-cell malignancies. Invivoscribe Inc. (San Diego, CA, USA) is currently performing the validation of their commercial assays to apply for final approval from FDA. ASOqPCR: allele-specific oligonucleotide quantitative PCR; BM: bone marrow; ddPCR: digital droplet PCR; MFC: multiparametric flow cytometry; NGF: next-generation flow; NGS: next-generation sequencing; SHM: somatic hypermutation.

	Standard MFC	NGF	ASOqPCR	NGS	ddPCR
Applicability	90–100%	90–100%	40–75%	~90%	Comparable to qPCR
Sensitivity	10^−4^–10^−5^	10^−5^–10^−6^	10^−4^–10^−5^	10^−5^–10^−6^	At least 10^−5^
Standardization	No	EuroFlow	EuroMRD	ClonoSEQ *	Ongoing
Turnaround time	1 day	1 day	≥1 week	4 days–1 week	≥1 week
Specific primers/probes	Not applicable	Not applicable	Yes	No	Yes
Standard curve	Not applicable	Not applicable	Yes	No	No
Influenced by SHM	No	No	Yes	Yes	Yes
Baseline BM	No	No	Yes	Yes	Yes
Fresh sample (processing time)	Yes (24–48 h)	Yes (24 h)	No	No	No

## Data Availability

Not applicable.

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
