# Peer review of "Minimal Residual Disease in Multiple Myeloma: Past, Present, and Future"

_cancers, 2023, doi:10.3390/cancers15143687_

Round 1

Reviewer 1 Report

Medina-Herrera et al wrote a review about  minimal residual disease (MRD) in multiple myeloma: past, present and future.  MRD in multiple myeloma is a hot topic. The methods used to measure the MD level are well described. The clinical implications to use MRD data are adequate. Pitfalls are identified and discussed. The view in the future is realistic. 

Author Response

We appreciate the comments and would like to thank the reviewer.

Reviewer 2 Report

The article titled "Minimal residual disease in multiple myeloma: Past, present, and future" by Alejandro Medina-Herrera et al. provides a comprehensive and insightful review of the importance of minimal residual disease (MRD) detection in multiple myeloma patients. The authors effectively highlight the advancements in treatment and improved response rates achieved with new drugs. The discussion on the limitations of current response criteria and the need for updated evaluation methods is well-founded.

The review covers a wide range of MRD detection approaches, including molecular and immunophenotypic methods, offering a balanced perspective on their advantages and potential pitfalls and  successfully emphasizes the clinical relevance of MRD, addressing its prognostic role and its impact on treatment strategies.

Overall, this review presents a concise and informative analysis of the current state-of-the-art on MRD detection in multiple myeloma. The author's critical perspective, clarity of writing, and inclusion of relevant literature make it a valuable contribution to the field. I recommend this article for publication after minor revisions:

Minor revisions:

1. I would suggest a differentiation between simple summary and abstract in the use of expression and the structure. In the current state several repetitions exist between the two.

2. The manuscript is well-written. However some sentences need some clarification. I add some line numbers for reference: 36, 86, 135-136

Fine. Minor errors

Reviewer 3 Report

please see enclosed

please see file

Author Response

The authors report on the significance of MRD in MM by using different techniques such as NGS, PCR or NGF. This paper is well written and provides a good overview on this approach in MM.

We thank the reviewer for their time and suggestions that have certainly improved the quality of the manuscript.

I have some comments:

Figure: is the tumor cell number shown in absolute values? The authors write that `the magnitude of reduction of the tumor burden is proportional to the time required for progression´. This statement need clarification. Do the authors have scientific data for this assumption? Does it mean that e.g. 50% reduction of clonal PC is association with a certain time to progression? And 90% reduction with a proportional longer time to progression? Does the authors refer to a relative or absolute change in both these values?

Several publications have shown that stratification of patients based on different MRD thresholds effectively discriminates progression-free survival rates (Martínez-López et al. 10.1182/blood-2014-01-550020; Perrot et al. 10.1182/blood-2018-06-858613, supplemental figure 1); Rawstron and colleagues (10.1182/blood-2014-07-590166) even found that each log reduction in MRD levels translated into ~1 year of survival benefit, but of course many other factors play a role on this matter ‒current and prior treatment regimens, other diagnostic tests and risk markers, as well as all of the signs, symptoms, co-morbidities, and quality-of-life factors‒ and that observation is certainly not strictly proportional.

We have amended the figure and the legend as suggested by reviewers 3 and 4, including a new reference (8).

Line 129: `CD27, CD45 and CD81 have low or negative expression; CD56, CD28 and CD117 are generally overexpressed´. Please check these statements. CD81 might be expressed by clonal plasma cells. CD56 is only positive in part of MM while the association with the prognosis is still not fully clear. CD117 is likewise only positive in part of the patients and associated with distinct disease characteristics.  

We have consequently modified the sentence as follows: “CD27, CD45 and CD81 generally have lower or negative expression compared to normal/reactive PC; CD56, CD28 and CD117 are overexpressed in 60–75%, 15–45% and 30–32% of cases, respectively”

Table 1. The MRD rate does not fit to the numbers in the table. I assume with author mean the rate of patients with MRD negativity and not the number of patients with MRD (please see also the following tables).

We appreciate this comment as that was indeed the rate of MRD negative patients. We have changed Tables 1 and 2 (now Tables 2 and 3) accordingly.

Table 1. The CASSIOPEIA trial was a two part trial with different randomization. May the authors show MRD rates for the other 2 arms?

We have now included part 2 of the trial showing randomization for maintenance with daratumumab or observation.

 Line 323: `82.5% of patients achieving sCR and 91.8% of MRD negative patients´. What is the authors explanation that MRD negativity was achieved in more patients than sCR?

Line 323-326: What is the authors´ explanation that CAR T cell therapy results in MRD negativity in >90% of patients while this is achieved in <30% of patients by teclistamab?

Responses were assessed in all patients treated with cilta-cel (N=97), while MRD was evaluated only in 61 of them.

We have noticed that those differences in MRD rates between studies, as we wrote them, are due to the way data is presented in each original paper: MRD negativity rates considering the intention-to-treat population are 57.7% (56/97) for cilta-cel and 26.7% (44/165) for teclistamab. Considering only those patients that could be evaluated, MRD rates are 91.8% (56/61) and 81.5% (44/54) for cilta-cel and teclistamab, respectively. We have modified this paragraph accordingly.

Having said that, there are a number of plausible explanations for the improved responses and MRD rates that have been achieved with cilta-cel:

1) Presence of a costimulatory domain increases the efficacy of CAR T cells. Previous constructs lacking this domain in first-generation CAR T cells were not as successful.

2) Patients treated with CAR T cells often receive bridging therapy, which sometimes reduces tumor burden. In the CARTITUDE-1, 73 patients received bridging therapy, and 33 of them experienced a reduction in their tumor burden. Lymphodepletion prior CAR T cell administration may also play a role.

3) The efficacy of teclistamab depends on its continuous administration. Adverse events causing the interruption of treatment (most especially in the form of infections) are frequent in patients treated with bispecific antibodies.

Table 3. Why is the turnaround time of NGF 2-3 days and not less?

We apologize for the error, as this was meant to be “2-3 hours”. However, to make it more realistic in the context of a routine laboratory, we have changed it to “1 day” as in standard MFC.

The authors might think to point out more in detail:

  • What is the significance of MRD negativity achieved by different drugs or drug combinations, e.g. does it make a difference if MRD negativity is achieved by drug X or Y?

Overall, answering to this question is virtually impossible in the current climate, as many more randomized trials focused on MRD testing should be completed in order to have more insight about how to interpret data. At the current time, we consider the combined analysis of ALCYONE and MAIA trials (https://doi.org/10.1182/blood.2020010439), as well as TOURMALINE-MM3 and -MM4 trials (10.1182/blood.2022016782), the most mature studies focusing on MRD interpretations.

We have extended section 4.b “What is the optimal time point for MRD evaluation?” in order to stress the main findings of both studies.

  • What role has the sequential MRD assessment? What is the prognostic difference between repeated MRD negativity vs. change from MRD negativity to MRD positivity vs. change from MRD positivity to its negativity?

The prognostic difference between sustained MRD negativity and MRD conversion (MRD- to MRD+ or backwards) has been mentioned in sections 2.a.ii (page 5, “Very recently, Benjamin Diamond and colleagues…”), 4.b (page 14), and with a final comment in section 5.

  • Do the authors recommend MRD assessment in the clinical routine? If yes, by which method? In which patients? At what time point?

We have included a final statement with recommendations about the use of MRD in the ‘Conclusions’ section.

There are few language errors (e.g. `immunophenotipic´ instead of `immunophenotypic´ in the abstract, line 170 `patients´ instead of `patents´). Please check abbreviations. NGS is explained twice, BM is not explained.

We have corrected misspelling and checked abbreviations.

Reviewer 4 Report

In the review about biomarkers for minimal residual disease in multiple myeloma written by Medina-Herrera et al.

current biomarkers in MM that go beyond age, cytogenetics and treatment response are discussed. Since treatment schedules containing

lenalidomide, bortezomib and dexamethasone have come of age, new treatments such as CD38 antibodies, CAR-T-cell therapies, ADC targeting BCMA, BCL2 inhibitors for t(11;14) patients, as well as XPO1 inhibitor Selinexor are being tested in clinical trials and will change the standards of MM treatment. Because this plethora of different treatment schedules is not likely to diminish in the future a standardized method to measure clinical response in this disease is indeed highly warranted.

The review is of interest and covers a topic that is really important. Despite this it is currently a bit difficult to read.

1. To improve, it would need some prioritization of which studies are more important than others. Also not merely say what has been done but say what in the future may be the solution or the better technique that is relatively easy to perform and still effective enough. Make your expert opinion more visible.

2. A the beginning (line 34) the section were existing treatment schedules 

and new options are being described should be a bit longer and more in depth. Explain the different drug types.

(along these lines: lenalidomide, bortezomib and dexamethasone…

New treatments: such as CD38 antibodies, CAR-T-cell therapies, ADC targeting BCMA, BCL2 inhibitors for t(11;14) patients, as well as XPO1 inhibitor Selinexor).

This would help the reader who not directly works with MM patients.

3. Add a table (in the region of line 50-70) of classical clinical markers that have been described with their impact on overall/5-year survival or progress, also add the source of the information in the table. You can also add some selected (most important) new markers.

4. Also a table of the most used flow protocols would be helpful

5. As one example of too much detail and not enough expert opinion is line 438 to 450.

Here it would suffice to say that in principle tumor heterogeneity can better be reflected in 

ctDNA samples as compared to BM biopsies. However, since they are assessing the same tumor a correlation of data from ctDNA and BM would be expected but only a few studies do actually compare these. One of the few studies that has done this (124) found a lack of correlation stressing the importance of further in depths studies…

6. In the imaging part (line 476-512) it should be discussed beside the 

advantages and disadvantages of MRI versus PET/CT whether the concept of minimal residual disease as we know it from flow cytometry or DNA sequencing can be applied here or whether it is something fundamentally different in this context, which I would believe.

7. Give more power/content/thought to the which strategy is best part (line 591-598).

Minor:

1. In Figure 1 please make labeling of the axis clear Tumor cells/ml??

Time in years ? make ticks for every year. Is the x axis closer to 5 Years or 20 years…

I do understand that it is rather conceptual but a rough time estimation can be given.

2. Describe at the beginning of your work that M-proteins are released monoclonal antibodies (or fragments thereof from the plasma cells) and that their detection is used a surrogate marker for abundance of plasma cells.

English is fine

Author Response

Referee 4

In the review about biomarkers for minimal residual disease in multiple myeloma written by Medina-Herrera et al. current biomarkers in MM that go beyond age, cytogenetics and treatment response are discussed. Since treatment schedules containing lenalidomide, bortezomib and dexamethasone have come of age, new treatments such as CD38 antibodies, CAR-T-cell therapies, ADC targeting BCMA, BCL2 inhibitors for t(11;14) patients, as well as XPO1 inhibitor Selinexor are being tested in clinical trials and will change the standards of MM treatment. Because this plethora of different treatment schedules is not likely to diminish in the future a standardized method to measure clinical response in this disease is indeed highly warranted.

The review is of interest and covers a topic that is really important. Despite this it is currently a bit difficult to read.

We would like to thank the reviewer for their thorough correction of the manuscript and suggestions made to improve its quality.

  1. To improve, it would need some prioritization of which studies are more important than others. Also not merely say what has been done but say what in the future may be the solution or the better technique that is relatively easy to perform and still effective enough. Make your expert opinion more visible.

Our aim was to provide an extensive, evidence-based review about the history of MRD in MM. Although some studies are certainly more relevant than others in introducing key notions for understanding the nature of MRD and its clinical utility, and some techniques mentioned appear to be less useful in the case of MM, we wanted to reflect in an unbiased way the work done in this field over the years.

Regarding the optimal method, we believe we had already addressed the issue in section 4.d. We have also included a final statement with our recommendations in section 5.

  1. A the beginning (line 34) the section were existing treatment schedules and new options are being described should be a bit longer and more in depth. Explain the different drug types. (along these lines: lenalidomide, bortezomib and dexamethasone…

New treatments: such as CD38 antibodies, CAR-T-cell therapies, ADC targeting BCMA, BCL2 inhibitors for t(11;14) patients, as well as XPO1 inhibitor Selinexor).

This would help the reader who not directly works with MM patients.

Following the recommendation, we have expanded the first paragraph to introduce most of the available drugs for MM, including a new reference.

  1. Add a table (in the region of line 50-70) of classical clinical markers that have been described with their impact on overall/5-year survival or progress, also add the source of the information in the table. You can also add some selected (most important) new markers.

 We have included a new Table (Table 1) with some of the classical and novel risk markers with their impact on prognosis.

  1. Also a table of the most used flow protocols would be helpful

We would like to keep outside of the manuscript detailed methodological aspects such as that proposed by the reviewer, given that we believe this is not the scope of the review. Moreover, the most objective and impartial solution would be including similar tables for all methods, which could make the article much longer.

  1. As one example of too much detail and not enough expert opinion is line 438 to 450.

Here it would suffice to say that in principle tumor heterogeneity can better be reflected in  ctDNA samples as compared to BM biopsies. However, since they are assessing the same tumor a correlation of data from ctDNA and BM would be expected but only a few studies do actually compare these. One of the few studies that has done this (124) found a lack of correlation stressing the importance of further in depths studies…

Following the suggestion, we have reduced section 3.a. making it more concise.

  1. In the imaging part (line 476-512) it should be discussed beside the advantages and disadvantages of MRI versus PET/CT whether the concept of minimal residual disease as we know it from flow cytometry or DNA sequencing can be applied here or whether it is something fundamentally different in this context, which I would believe.

 We appreciate the suggestion as it is in fact an important aspect to mention. Therefore, we have included a sentence in the first paragraph of section 3.b: “Since imaging captures the expansion of tumor cells at a whole-body level, distinguishing active from inactive lesions, perhaps it should be interpreted as a second layer of MRD.”

  1. Give more power/content/thought to the which strategy is best part (line 591-598).

 We have provided additional thoughts about the optimal strategy in section 5.

Minor: 

  1. In Figure 1 please make labeling of the axis clear Tumor cells/ml??

Time in years ? make ticks for every year. Is the x axis closer to 5 Years or 20 years…

I do understand that it is rather conceptual but a rough time estimation can be given.

We have amended the figure and the legend as suggested by reviewers 3 and 4, including a new reference (8).

  1. Describe at the beginning of your work that M-proteins are released monoclonal antibodies (or fragments thereof from the plasma cells) and that their detection is used a surrogate marker for abundance of plasma cells.

We have introduced the notion at the beginning of section 1.c.

Round 2

Reviewer 3 Report

thank you. I have no further comments

Reviewer 4 Report

The performed changes have greatly improved the manuscript.

I would accept it in the present form.